# Macrophage Long Non-Coding RNAs in Pathogenesis of Cardiovascular Disease

**DOI:** 10.3390/ncrna6030028

**Published:** 2020-07-11

**Authors:** Marcin Wysoczynski, Jae Kim, Joseph B. Moore, Shizuka Uchida

**Affiliations:** 1Diabetes & Obesity Center, Department of Medicine, The Christina Lee Brown Envirome Institute, University of Louisville School of Medicine, Louisville, KY 40202, USA; jae.kim@louisville.edu (J.K.); joseph.moore@louisville.edu (J.B.M.I.); 2Cardiovascular Innovation Institute, School of Medicine, University of Louisville, Louisville, KY 40202, USA

**Keywords:** cardiovascular disease, inflammation, lncRNA, macrophage

## Abstract

Chronic inflammation is inextricably linked to cardiovascular disease (CVD). Macrophages themselves play important roles in atherosclerosis, as well as acute and chronic heart failure. Although the role of macrophages in CVD pathophysiology is well-recognized, little is known regarding the precise mechanisms influencing their function in these contexts. Long non-coding RNAs (lncRNAs) have emerged as significant regulators of macrophage function; as such, there is rising interest in understanding how these nucleic acids influence macrophage signaling, cell fate decisions, and activity in health and disease. In this review, we summarize current knowledge regarding lncRNAs in directing various aspects of macrophage function in CVD. These include foam cell formation, Toll-like receptor (TLR) and NF-kβ signaling, and macrophage phenotype switching. This review will provide a comprehensive understanding concerning previous, ongoing, and future studies of lncRNAs in macrophage functions and their importance in CVD.

## 1. Introduction

Macrophages are a subtype of immune cells that comprise an important fraction of the innate immune system. Initially described by Ilia Metchnikoff in 1882, the macrophage was identified as a cell capable of ingesting particles and/or other cells in a process termed phagocytosis—a function heralded as a protective mechanism against pathogens [1]. Since then, our knowledge regarding the role of macrophages in human physiology and pathology has substantially evolved [2]. Macrophages are actively recruited to various tissues during early development to facilitate organ development and function [3,4]. Tissue-resident macrophages are deposited in two independent waves during embryonic development. First, a transient hematopoietic wave of erythro-myeloid progenitors (EMPs) emerges from the posterior plate mesoderm and forms yolk sac blood islands. These yolk sac-derived macrophages are formed independently of hematopoietic stem/progenitor cells and monocytes. They are called “primitive macrophages” due to their immature immunophenotype (i.e., absence of typical macrophage markers) and lack of detectable phagocytic activity. The second wave starts when EMPs seed fetal liver and differentiate to fetal monocytes and further form macrophages. Thus, tissue-resident macrophages deposited during embryogenesis are composed of yolk sac “primitive macrophages” and fetal liver monocyte-derived macrophages [4,5,6,7]. 

Many tissue macrophages deposited in organs during development remain in adult tissues. These tissue-resident macrophages have a distinct phenotype and function from monocyte-derived macrophages, which are recruited to tissues and organs during injury and/or infection [4,8]. Tissue-resident macrophages are highly abundant in all organs, including the skin, gastrointestinal tract, lymphoid organs, and lungs. Macrophage populations are aptly named according to the tissue in which they reside, as well as their function; for example, Kupffer cells refer to macrophages in the liver, microglia in the brain, and alveolar macrophages in the lung [4,9]. In general, all macrophages share several important functions: (1) they play a role in tissue development and maintenance; (2) tissue surveillance and sampling; and (3) pathogen clearance, antigen presentation, inflammation resolution, and tissue repair. Depending on the organ or tissue localization, macrophages may fulfill several of these tasks and, as such, comprise highly heterogeneous cell populations with various phenotypes and functions [9]. Macrophages are also known for their plasticity, as they may undergo phenotype switching in response to various stimuli [4,10]. For example, lipopolysaccharide (LPS) exposure stimulates macrophage activation and the production of proinflammatory cytokines, while interleukin 4 (IL-4) stimulation elicits anti-inflammatory cytokine secretion [11]. Although many signaling pathways have been identified that direct macrophage polarization and function, it is not fully understood how such pathways are regulated; however, RNA-mediators, such as long non-coding RNAs (lncRNAs), have emerged as important signaling components dictating various aspects of macrophage biology [12,13,14]. To this end, here, we review the current knowledge regarding the regulation of macrophage function by lncRNAs in the context of cardiovascular disease (CVD). Compared to other similar review articles in recent years [15,16], we summarize current knowledge regarding lncRNAs in directing various aspects of macrophage functions in CVD, including foam cell formation, Toll-like receptor (TLR) and NF-kβ signaling, and macrophage phenotype switching.

## 2. Role of Macrophage in Cardiovascular Disease

CVD broadly describes a class of diseases that affects the heart and/or blood vessels. A host of risk factors (including high blood pressure/hypertension, smoking, diabetes and obesity, and sedentary lifestyle) have been identified as major contributors to CVD. Such risk factors contribute to the accumulation of fats, cholesterol, and inflammatory cells in coronary vasculature, which can lead to plaque formation and blood flow restriction—a pathology commonly referred to as atherosclerosis. The role of macrophages in the pathogenesis of atherosclerosis has been well-established. Much of what is now known regarding macrophages in this process is derived from preclinical animal models of atherosclerosis and clinical cohorts [17,18,19,20]. Animal models include mice deficient in apolipoprotein E (ApoE) supplemented with a Western-style diet. In these models, elevated fat and sugar levels trigger an emergency-mediated hematopoietic response in the bone marrow and spleen, resulting in the overproduction of monocytes [19,21,22]. The surplus of splenic- and bone marrow-derived monocytes is released into the circulation, where they adhere to the atherosclerotic endothelium and extravasate into lesions, where they differentiate into macrophages. The combinatorial accumulation of monocyte-derived macrophages and lipids in the perivascular space leads to the retention of lipid droplets within the macrophage’s cytoplasm, giving them the characteristic appearance of foam-like structures. The accumulation of foam cells in arterial walls is a hallmark of early atherosclerotic lesion formations [19,20]. An unremitting accumulation of monocytes and their lineage descendant macrophages can contribute to fibrous cap thickening, hematoma, thrombi, calcification, and the degeneration of plaque integrity. The relative abundance of macrophages in atherosclerotic plaques is regulated by their exit and/or death but, also, the sustained recruitment of monocytes. An efficient macrophage exit, or reduced monocyte recruitment, results in a reduction in the lesional macrophage number and regression of the disease. On the other hand, factors contributing to increased monocyte recruitment accelerate atherosclerosis [23,24]. 

The destabilization and rupture of arterial plaque may lead to partial or complete occlusion of the coronary artery, which can lead to acute cardiac ischemic events, such as myocardial infarction (MI). Ischemic injury to the myocardium triggers a sterile systemic inflammatory response that is required for the activation of the tissue-healing program [25,26,27]. In response to MI, the bone marrow and spleen generate a surplus of myeloid cells, mostly neutrophils and monocytes, which are released to the peripheral blood and, subsequently, home to the damaged myocardium [28,29]. Recruited neutrophils and monocytes are necessary for the removal of necrotic tissue, the initialization of angiogenesis, and the stimulation of myofibroblasts for collagen synthesis and wound healing [27]. This initial myocardial myeloid response is imperative for proper healing. A reduction of neutrophil or monocyte/macrophage infiltration, either through systemic depletion or splenectomy, impairs optimal post-MI healing and results in progressive myocyte death, excessive fibrosis, and possible cardiac rupture [29,30]. On the other hand, excess neutrophils and monocyte/macrophage recruitments lead to an abundant secretion of proinflammatory mediators that negatively affect tissue healing and contribute to myocardial damage [31]. Thus, a balanced immune response due to ischemic injury is necessary for proper infarct healing and recovery after MI. 

On the other hand, despite the apparent beneficial effects of myeloid cells on infarct healing, macrophages also contribute to the progression of fibrosis and impaired pump functions in chronic heart failure. Following MI, the heart experiences significant cell death and tissue necrosis. The myocardium is then rapidly overwhelmed by an intense inflammatory response that facilitates both the removal of dead/necrotic tissue, as well as supports tissue replacement with an akinetic, collagen-based scar [25,27]. This remodeling process is an adaptive response that serves to preserve the structural integrity of the myocardium after ischemia-mediated tissue loss. However, the remodeling process, characterized by ongoing collagen deposition (defined as reactive fibrosis) and reduced ventricular compliance, is considered maladaptive in the long term—contributing to sustained myocyte dropout, cardiac hypertrophy, and progressive decrements in ventricular function, which eventually culminate in end-stage heart failure [32]. Systemic inflammation is considered a hallmark feature of chronic heart failure in patients [25,26,27]. Proinflammatory cytokine levels, such as tumor necrosis factor-alpha (TNF-α) and IL-6, are closely associated with the heart failure status, suggesting that inflammatory cytokine signaling contributes to progressive pump failure. Likewise, studies in mice models of heart failure demonstrate that, after MI, there is a sequential (4-8 weeks after MI events) accumulation of macrophages in nonischemic regions of the heart, which may provoke continued matrix deposition, myocardial stiffening, and impaired ventricular performance [32,33]. Compatible with this view, the inhibition of monocyte infiltration via the blockade of adhesion molecules, or splenectomy in mice with established heart failure, dampens and improves adverse cardiac remodeling [32,33]. These findings highlight a potentially detrimental role for macrophages in chronic heart failure. While the precise molecular mechanisms underlying macrophage development, polarization, and biologic activity/function in atherosclerosis and heart failure pathophysiology remain a focus of intense scientific investigation, much remains unknown. Nevertheless, recent studies have brought attention to the importance of lncRNAs in macrophage biology—providing evidence that they actively participate in gene regulatory networks during complex biological processes.

## 3. Non-Coding RNA Nomenclature

Though lncRNAs are broadly defined as any non-protein-coding RNAs whose lengths are longer than 200 nucleotides, they are ultimately stratified according to their genomic position in relation to nearby protein-coding genes (i.e., messenger RNAs (mRNAs)). If a lncRNA is located on the same genomic DNA strand as a nearby protein-coding gene, it is defined as a sense lncRNA. In many instances, a sense lncRNA may overlap with an adjacent protein-coding gene; these lncRNAs are referred to as sense-overlapping lncRNAs. If a lncRNA is located on an opposite genomic DNA strand relative to a neighboring protein-coding gene, it is defined as an antisense lncRNA. Both sense and antisense lncRNAs are commonly named after their nearest protein-coding genes [34]. A lncRNA can also arise from intronic sequences of protein-coding genes, which are aptly called intronic lncRNA. Furthermore, a lncRNA can be distally located far from, but between, two protein-coding genes. Such types of lncRNAs are defined as long intergenic lncRNAs (lincRNAs). When lincRNAs are located in the enhancer or promoter region of a protein-coding gene, these lncRNAs are defined as enhancer- or promoter-associated lincRNAs (eRNAs and pRNAs, respectively). More recently, a new type of lncRNA was defined, which are called circular RNAs (circRNAs). These derive from splicing events and possess stable circular conformations. Although lncRNAs are subdivided based on their genomic locations and in conjunction with nearby protein-coding genes, the definition of non-protein-coding genes remains contested, as some lncRNAs encode micropeptides, as in the case of *Myoregulin* (*MLN*) [35]. Furthermore, pseudogenes, whose genomic sequences are similar to functional protein-coding genes, should be included in the category of lncRNAs, as pseudogenes do not encode for proteins [36].

## 4. Mechanism of lncRNA Function

As many lncRNAs are not known to possess any intrinsic catalytic activity themselves, it is believed that lncRNAs influence biological processes by interacting with macromolecules (e.g., DNA, RNA, and proteins). Since the number of lncRNAs well-surpasses that of protein-coding genes, the categorization of lncRNA functions is challenging. Nevertheless, four broad categories of lncRNA functions were defined. The first category is genomic imprinting, which is an epigenetic phenomenon whereby genes are expressed in a parent-of-origin-dependent manner [37]. Such imprinting lncRNAs include the antisense of IGF2R non-protein coding RNA (*Airn*) [38], H19-imprinted maternally expressed transcript (*H19*) [39,40], maternally expressed 3 (*MEG3*) [41], maternally expressed 8, small nucleolar RNA host gene (*MEG8*) [42], and X-inactive specific transcript (*XIST*) [43]. The second category of lncRNA functions is transcription regulation. LncRNAs can directly bind to genomic sequences and form a scaffold for the recruitment of epigenetic and/or transcriptional factors, which influence transcriptional activity [44]. This functional category has been investigated extensively in recent years, as many binding partners (i.e., epigenetic and transcription factors) have been studied in the field of epigenetics. Furthermore, the methods to investigate such mechanisms (i.e., chromatin immunoprecipitation followed by sequencing (ChIP-seq)) are well-established, which makes it easier to elucidate the function of lncRNAs in this context. However, several laboratories have reported the promiscuity of RNA binding of epigenetic factors (especially, enhancer of the zeste homolog 2 (EZH2), a catalytic component of the polycomb repressive complex 2 (PRC2) [45,46,47,48]), suggesting that caution be taken when interpreting lncRNAs as transcriptional regulators. The third category of lncRNA functions is that of post-transcriptional regulation, where their complementation with mRNAs may influence RNA stability. Further, one should note that some lncRNAs encode microRNAs (miRNAs) themselves in their genomic sequences, and others can bind miRNAs and function as miRNA sponges (also known as competing endogenous RNA (CeRNA)) [49]. This is an attractive mechanism to control the availability of miRNAs, as miRNAs bind to the 3’-untranslated regions (3’-UTR) of target mRNAs to induce mRNA degradation and translational repression. However, some lncRNAs (e.g., tumor protein p53 pathway corepressor 1 (*TRP53COR1*, also known as *lincRNA-p21*), HOX transcript antisense RNA (*HOTAIR*), and metastasis-associated lung adenocarcinoma transcript 1 (*MALAT1*)) are degraded by miRNAs rather than functioning as miRNA sponges [50]; thus, a thorough investigation of the interactions between lncRNAs and miRNAs is necessary. The last category of lncRNA functions is translation control. Through binding to RNA-binding proteins (RBPs), lncRNAs can influence the translational efficiency of mRNAs. As more and more RBPs are being identified from high-throughput screening technologies [51,52,53], there is a growing number of lncRNAs identified as RBP sponges, including those of circRNAs [54]. Thus, just as there exist multifunctional proteins, many lncRNAs, too, exhibit versatility in terms of function—which highlights a likely complex and multidimensional role for lncRNA signaling in immune cell biology.

## 5. Role of Macrophage lncRNA in Foam Cell Formation

The hallmark feature of atherosclerosis is the formation of atherosclerotic plaques in the intimal layer of the artery [55,56]. At the early stages of atherosclerosis, low-density lipoproteins (LDLs) pass through arterial endothelial junctions and become retained in the subendothelial space via interactions with matrix proteoglycans. Trapped LDLs aggregate and become oxidized to form oxidized low-density lipoproteins (ox-LDLs), which divert their binding from LDL receptors to scavenger receptors (SRs) expressed on monocytes and their descendant macrophages. This interaction ultimately results in the recruitment of monocyte/macrophage populations to the intima of the arterial wall [57]. These SRs include CTD-associated factor 1 (SCAF1, also known as SR-A1), cluster of differentiation 36 (CD36), and lectin-like ox-LDL receptor-1 (LOX-1), which are upregulated after stimulation with ox-LDLs or pro-inflammatory stimuli [58]. After binding to macrophage SRs, ox-LDLs are internalized and transported to lysosomes. There, lipoprotein cholesteryl esters (CE) are hydrolyzed by lysosomal acid lipase (LAL) to generate free cholesterol (FC) [59]. FC are then trafficked from lysosomes through the coordinated action of Niemann-Pick Type C1 and 2 (NPC1 and 2) membrane proteins and, eventually, transported to endoplasmic reticulum (ER), where they become converted back to CE by Acyl-CoA cholesterol acyltransferase (ACAT) through a process of re-esterification. FC can also be transported out of cells by ATP-binding cassette transporters A1 (ABCA1) to form new high-density lipoprotein (HDL) particles [60]. Excess accumulation of CE in lysosomes and the ER leads to the formation of lipid droplets in the macrophage with the appearance of foam-like structures, commonly referred to as macrophage-derived foam cells (Figure 1) [55,56]. Further, CE and FC accumulation leads to the inhibition of LAL and ABCA1 activity, which is believed to contribute to the pathogenesis of atherosclerosis. The main cells involved in the process of foam cell formation include smooth muscle cells, endothelial cells, and macrophages [55,56]. Several methods have been used to identify lncRNAs involved in the pathogenesis of atherosclerosis. Patient-sourced samples (including atherosclerotic plaques and blood samples), as well as animal models of atherosclerosis, have facilitated the identification of numerous candidate lncRNAs (discussed in 5.1-3) [61,62,63]. Additionally, large numbers of lncRNAs involved in atherosclerosis were identified during foam cell formations in vitro, an approach that effectively recapitulates the accumulation of lipid droplets and acquisition of a macrophage foam cell-like phenotype through the incubation of macrophages with ox-LDL [62,64].

### 5.1. Cholesterol Update—Regulation of Scavenger Receptor Expression

The first step in forming macrophage-derived foam cells involves cholesterol uptake. Three key scavenger receptors were described in macrophages to bind and internalize ox-LDL. As mentioned previously, these are CD36, SR-A1, and LOX-1 [55,56]. CD36 belongs to the scavenger receptor class B family. It consists of an extracellular loop domain with two transmembrane and two cytoplasmic domains. CD36 is a high-affinity ox-LDL receptor that is highly expressed in macrophages [65,66]. The screening utilizing ox-LDL-stimulated macrophages in vitro has resulted in the identification of three candidates lncRNAs that regulate CD36 receptor expression via acting as miRNA sponges. Hu et al. [67] found that the lncRNA urothelial cancer-associated 1 (*UCA1*) was upregulated in the human monocytic cell line (THP-1 macrophages) after stimulation with ox-LDLs. *UCA1* knockdown inhibited the expression of CD36 and limited foam cell formation in response to ox-LDL. The overexpression of *miR-206*, identified as a predicted target of *UCA1*, decreased oxidative stress induced by ox-LDL and was reversed by *UCA1* upregulation in vitro, suggesting that *UCA1* regulates CD36 expression via *miR-206* sequestration (“sponging”) [67]. In an analogous study, Wang et al. [68] found that ox-LDL-stimulated THP-1 macrophages yielded an increased expression of the lncRNA nuclear paraspeckle assembly transcript 1 (*NEAT1*) and a decreased expression of *miR342-3p*. The inhibition of *NEAT1* in THP-1 macrophages stimulated with ox-LDL repressed the expression of CD36 and cholesterol accumulation. Additionally, a decreased expression of *NEAT1* led to repression of the inflammatory gene expression program (cytochrome c oxidase subunit II (COX2), interleukin 6 (IL6), interleukin 1 beta (IL1B, also known as IL-1β), and tumor necrosis factor (TNF, also known as TNF-α)) in vitro. Bioinformatics screening identified *miR342-3p* as a direct target of *NEAT1*, indicating its potential sponging activity. In further studies, the overexpression of *miR-342-3p* impaired ox-LDL-induced foam cell formation through the downregulation of CD36 expression [68]. Independently, Chen et al. [69] found that *NEAT1* is upregulated in the mouse RAW264 macrophage cell line stimulated with ox-LDLs, which coincided with an upregulation of CD36. Mechanistically, *NEAT1* was shown to sponge *miR-128*, which led to the upregulation of CD36. On the contrary, *NEAT1* knockdown restored the expression of *miR-128* and reduced the expression of CD36 in response to ox-LDL stimulation [69]. In a similar study, Liu et al. [70] found the lncRNA *HOTAIR* to regulate CD36 expression, cholesterol uptake, reactive oxygen species (ROS) production, and proinflammatory transcriptional program activation in THP-1 cells stimulated with ox-LDL via the sponging of *miR330-5p*. These studies show that the expression of CD36 is subject to significant lncRNA-mediated regulation and further provides evidence that this mechanism involves the lncRNA-mediated sequestration of numerous downstream miRNA targets; precisely how those key downstream miRNA targets regulate CD36 expression was not addressed [70].

Differential mechanisms for the lncRNA-mediated regulation of CD36 expression were noted in a study by Huangfu et al. [71]. There, the lncRNA, *MALAT1*, was identified as a binding partner of β-catenin. An increased *MALAT1* expression in response to ox-LDL led to an enhanced β-catenin binding to the CD36 receptor promoter, which was reversed by the inhibition of *MALAT1* [71]. In another study, which employed an RNA sequencing-based approach, identified *LINC01272* as a highly expressed lncRNA in patient-derived atherosclerotic plaques [67]. As *LINC01272* was markedly upregulated in unstable plaques, it was aptly named the “plaque-enriched lncRNA in atherosclerotic and inflammatory bowel macrophage regulation” (*PELATON*). *PELATON* was notably enriched in macrophages and exclusively localized in the nuclear fraction, implicating a likely role in the regulation of gene transcription. As anticipated, in vitro experiments demonstrated that *PELATON* knockdown effectually impaired macrophage phagocytosis, lipid uptake, and reactive oxygen species production, as well as CD36 expression. Detailed mechanistic studies providing greater insights into the *PELATON*-mediated regulation of CD36 were not conducted [72]. Regardless of the many advancements in RNA biology to date, no known studies are investigating the impact of lncRNAs on those other receptors (e.g., SR-A1 and LOX-1) in influencing foam cell formation. However, recent studies have begun to focus on the contributions of endothelial and smooth muscle cells to foam cell formation from the perspective of SR-A1 and LOX-1-dependent signaling [73,74,75].

### 5.2. Reverse Cholesterol Transport – ABCA1 and ABCG1 Regulation

In addition to cholesterol uptake by scavenger receptors, the balance of FC and CE is also a critical factor in the regulation of cholesterol content in macrophage-derived foam cells. In homeostasis, after internalization, lipoproteins are delivered to the late endosome, where CE is hydrolyzed into FC by lysosomal acid lipase (LAL). Subsequently, FC is transported to the ER, where it is re-esterified by acetyl-CoA acetyltransferase 1 (ACAT1) and then stored in the form of lipid droplets [55,56]. When the flux of FC from lysosomes is high, oxysterol levels, including 27-hydroxycholesterol, are increased and bind to the nuclear liver X receptor (LXR). This interaction leads to the activation of a transcriptional program that directs the removal of cholesterol from cells via the initiation of reverse cholesterol transport [76]. One of these includes the ATP-binding cassette transporter ATP-binding cassette subfamily A member 1 (ABCA1). This promotes the cholesterol efflux to Apolipoprotein A1 (APOA1), which is the rate-limiting step in the formation of HDL. In addition to ABCA1, two other transporters are involved in reverse cholesterol transport: ATP-binding cassette subfamily G member 1 (ABCG1) and scavenger receptor class B member 1 (SCARB1, also known as SR-BI) [60,62,64]. During the initial steps of atherosclerosis, an increased uptake of cholesterol and defects in cholesterol efflux ultimately results in the accumulation of lipid droplets and, in turn, foam cell formation [55,56].

ABCA1 is a member of the larger superfamily of ABC transporters. It is well-established that ABCA1 plays a critical role in the prevention of macrophage foam cell formation. ABCA1 mutants yield reduced HDL cholesterol levels and heightened atherosclerotic burdens compared with the controls [77]. The expression of ABCA1 is highly regulated at the transcriptional and post-transcriptional levels. At the transcriptional level, several nuclear receptors, including peroxisome proliferator-activated receptors (PPARs), liver X-receptor (LXR), and farnesoid X receptor (FXR), influence ABCA1 expression. Moreover, at the post-transcriptional level, the 3′-UTR of the *ABCA1* gene is directly targeted by multiple miRNAs, including *miR-33*, *miR-758*, *miR-145*, *miR-27*, *miR-144*, *miR-26*, and *miR-106*, to inhibit the cholesterol efflux [78,79]. Besides the above microRNAs, recent observations below suggest that lncRNAs play an active role in the dysregulation of cholesterol efflux from macrophages and foam cell formation.

Studies from three independent laboratories reported that long intergenic non-protein-coding RNA 1228 (*LINC01228*, also known as *DYNLRB2-2*) is upregulated in macrophages in response to ox-LDL. This lncRNA was associated with an enhanced cholesterol efflux and reduced macrophage-derived foam cell formation—findings that were ultimately shown to involve three distinct mechanisms [80,81,82]. In the study by Li Y et al. [80], THP-1 macrophages stimulated with ox-LDL exhibited elevated expression of *DYNLRB2-2*; the lncRNA promoted cholesterol efflux and inhibited foam cell formation through the activation of autophagy. Mechanistically, *DYNLRB2-2* modulated the *miR-298*/Sirtuin 3 (SIRT3) axis, which subsequently resulted in the LKB1/AMPK/mTOR signaling pathway-mediated initiation of an autophagic program in macrophages [80]. This mechanism supports previous observations indicating that autophagy (in this case, termed “lipophagy”) is a key mechanism of CE reverse transport from lipid droplets to lysosomes and the eventual ABCA1 transporter-mediated efflux [83]. Consistent with this idea, Li Y et al. [81] also provided evidence that elevated *DYNLRB2-2* in ox-LDL-stimulated THP-1 and RAW264.7 macrophages is associated with alterations in the ABCA1 expression and cholesterol efflux in vitro. Specifically, *DYNLRB2-2* overexpression inhibited the macrophage-derived foam cell formation, a finding that was accompanied by an enhanced cholesterol efflux and ABCA1 expression. Further, this study indicated that *DYNLRB2-2* negatively regulated the Toll-like receptor 2 (*TLR2*) expression. The TLR2 overexpression reversed the effects of *DYNLRB2-2* on the cholesterol efflux and ABCA1 expression levels in THP-1 and RAW264.7 macrophages [81]. These data are consistent with reports that macrophage TLR-2 is a protagonist of foam cell formation and atherosclerosis [84]. In another study [82], Hu et al. revealed that the *DYNLRB2-2*-mediated upregulation of ABCA1 in ox-LDL-stimulated THP-1 macrophages requires G protein-coupled receptor 119 (GPR119). Supporting this, the in vitro overexpression of GRP119 increased the cholesterol efflux, inhibited foam cell formation, and activated a proinflammatory genetic program. Furthermore, the in vivo viral overexpression in high-fat fed ApoE-/- mice (a murine model of atherosclerosis) showed that GRP119 has a protective effect against atherosclerosis via increasing the cholesterol efflux and reducing the proinflammatory cytokine expression. These studies ultimately suggest that *DYNLRB2-2* is a promising therapeutically exploitable target to increase cholesterol homeostasis and reduce atherosclerotic plaque formation. 

Another lncRNA, termed the “cholesterol-induced regulator of metabolism RNA” (*CHROME*), was demonstrated to be an imperative component of proper cholesterol homeostasis. Hennessy et al. [85] identified *CHROME* as elevated in the plasma and atherosclerotic plaques of individuals with coronary artery disease. *CHROME* expression is influenced by dietary and cellular cholesterol via the sterol-activated liver X receptor transcription factors, which control gene-mediating responses to cholesterol overloads. Using gain- and loss-of-function approaches, the authors show that *CHROME* promotes cholesterol efflux and foam cell formation, most likely through the regulation of miRNA expression. *CHROME* knockdown in human macrophages increases the levels of *miR-27b*, *miR-33a*, *miR-33b*, and *miR-128*, thereby reducing the expressions of their overlapping target gene networks, including *ABCA1*. Since *CHROME* is also expressed in hepatocytes, it can play an important role in systemic cholesterol homeostasis in atherosclerosis [85].

Macrophage-expressed lncRNAs can also have detrimental effects on foam cell formation through the inhibition of ABCA1. In the study by Meng et al. [86], the lncRNA growth arrest-specific 5 (*GAS5*) was highly expressed in the THP-1 macrophage-derived foam cells and localized primarily to the nucleus. The overexpression of *GAS5* facilitated lipid accumulation and foam cell formation. Further studies showed that *GAS5* inhibits ABCA1 by binding to the histone methyltransferase EZH2. The in vitro overexpression of EZH2 reduced the cholesterol efflux and facilitated lipid accumulation. ApoE-/- mice with overexpression of *GAS5* or EZH2 showed increased total cholesterol, free cholesterol, cholesterol ester, low-density lipoprotein levels, aortic plaque, and lipid accumulation, accompanied by reduced HDL levels and cholesterol outflow. These data suggest *GAS5* as a promising target to restore cholesterol homeostasis and inhibit atherosclerotic plaque formation in patients [86].

Numerous studies show that ABCG1 and SR-BI are also key components of cholesterol homeostasis in macrophages. In a recent study by Xu et al. [87], authors show that THP-1 macrophages and vascular smooth muscle cells (VSMC) stimulated with ox-LDL express a novel lncRNA, *AC096664.3*. Further studies in VSMC demonstrated that *AC096664.3* regulates the expression of ABCA1 via the inhibition of PPAR-γ expression. However, these findings were not confirmed in macrophages. As of late, no lncRNAs have been found to regulate scavenger receptor class B type 1 (SCARB1, also known as SR-BI).

## 6. Role of lncRNA in the Regulation of Macrophage Polarization

In addition to foam cell formation, macrophages may also contribute to atherosclerosis pathophysiology via the secretion of a broad spectrum of pro- and anti-inflammatory cytokines. Monocytes recruited to the sites of inflammation (e.g., atherosclerotic plaque or infarcted heart) differentiate to macrophages, which engage in shaping the inflammatory response through the secretion of cytokines, chemokines, and reactive oxygen species (ROS) [19,20]. Depending on the inflammatory milieu, tissue-resident macrophages can acquire a pro-inflammatory (M1) or anti-inflammatory/pro-resolving phenotype (M2) [4,10]. Naïve macrophage exposure to danger-associated molecular patterns (DMAPs), oxidized low-density lipoproteins (ox-LDLs), or interferon-gamma (IFN-γ) initiate the activation of an M1 phenotype, which leads to the secretion of proinflammatory cytokines IL-1, IL-6, tumor necrosis factor-alpha (TNF-α), IL-6, and interleukin 12 (IL-12) [57,88]. These cytokines perpetuate non-resolving inflammation and contribute to the pathogenesis of CVD, including atherosclerosis and heart failure. Toll-like receptor (TLR) stimulation leads to the activation of signaling pathways, culminating in proinflammatory gene activation—a chief characteristic of the M1 phenotype (Figure 2) [57,89]. Alternatively, stimulation with interleukin 4 (IL-4) and interleukin 13 (IL-13), as well as exposure to apoptotic cells and the activation of engulfment through TAM receptors (i.e., Tyro3, Axl, and Mer), can activate an anti-inflammatory and pro-resolving M2 phenotype in macrophages [57,89]. This results in the secretion of anti-inflammatory cytokines (IL-10 and TGFβ) but, also, pro-resolving lipid mediators (e.g., resolvins, maresins, and lipoxins), which are associated with the pro-reparative and anti-inflammatory actions of macrophages [90,91,92]. Recent literature suggests that lncRNAs are involved in the modulation of the macrophage phenotype, hence their likely involvement in the pathogenesis of CVD.

### 6.1. The Role of TLR Signaling in Macrophage Polarization 

TLRs are the most extensively studied and characterized pattern-recognition receptors involved in CVD. To date, 13 TLRs have been discovered and characterized. They can be broadly divided into two categories: cell membrane TLRs (TLR1, TLR2, TLR4, TLR5, TLR6, and TLR11) and cytoplasmic TLRs residing in ER, endosomes, and lysosomes, which are nucleic acid-sensing receptors (TLR3, TLR7, TLR8, TLR9, and TLR13). Recognition of the ligands results in TLR dimerization that activates intracellular Toll/interleukin-1 receptor/resistance protein (TIR) domains to enable the recruitment of adapter proteins like myeloid differentiation primary response 88 (MyD88), TIR-domain-containing adapter-inducing interferon-β (TRIF), transforming growth factor-β-activated kinase 1 (TAK1; official gene name, mitogen-activated protein kinase kinase kinase 7 (MAP3K7)), and mal T cell differentiation protein (MAL) [93,94,95,96]. All TLRs, except for TLR3, signal via the adapter protein MyD88, which recruits interleukin-1 receptor-associated kinases (IRAKs) to activate nuclear factor-κB (NF-κB)-dependent proinflammatory cytokines (all TLRs) and IRF7-dependent type I IFN (TLR7-9). TLR4 activates the MyD88-dependent pathway through the activation of the endosomal TIR domain-containing adapter-inducing interferon-β (TRIF)-dependent pathway to activate the expression of type 1 IFNs. TLR3 exclusively uses the TRIF-dependent pathway for downstream signaling. The activation of TLR signaling is associated with M1 macrophage phenotype and pro-inflammatory cytokine profiles (Figure 2) [93,94,95,96]. Thus, it will have a detrimental effect on the progression of CVD.

The participation of the TLR signaling pathways in the pathogenesis of CVD is well-studied in both CVD patients and small animal models [94]. It has been recognized that the pattern of TLR expression changes in CVD patients [97]. Studies in knockout animals have established that TLR signaling plays an active role in the pathogenesis of both atherosclerosis and heart failure [94,97]. Mice with a deficiency of TLR2 and TLR4 or downstream signaling proteins (i.e., IRAK4, TRAF6, TRIF, or MYD88) shows protection from atherosclerosis and various models of heart failure [94,96,98,99]. Therefore, understanding the molecular mechanisms regulating TLR signaling could lead to the discovery of new therapeutic targets. Since TLR signaling is regulated at multiple levels (including TLR expression) and epigenetic and post-translational mechanisms, it is possibly influenced by lncRNAs.

### 6.2. LncRNA Involved in the Macrophage TLR Activation and Signaling

The growing interest in lncRNAs has prompted the identification of lncRNAs that are involved in TLR signaling. For example, the lncRNA *AS-IL1a*, encoded within the IL-1α locus, is required for the recruitment of RNA polymerase II to the IL-1α promoter for its transcription in mouse primary bone marrow-derived macrophages [100]. Another example is *lnc-MARCKS*/*ROCKI* (Regulator of Cytokines and Inflammation), which interacts with the multifunctional DNA repair enzyme, apurinic/apyrimidinic endodeoxyribonuclease 1 (APEX1), to form a ribonucleoprotein complex and recruit histone deacetylase 1 (HDAC1) to remove the active enhancer mark, histone H3 acetylated at lysine 27 (H3K27ac). The removal of the H3K27ac mark results in the reduced expression of the most prominent cellular substrate for protein kinase C, myristoylated alanine-rich protein kinase C substrate (*MARCKS*), thereby regulating the expression of proinflammatory cytokines in macrophages [101].

To date, most loss-of-function experiments employed to elucidate the functions of lncRNAs are based on the usage of knockdown methods (e.g., antisense oligos (ASO), shRNAs, and siRNAs). As it is well-known that such acute silencing is associated with technical limitations, including off-target effects [102], the gold standard in developmental biology is to use knockout mice [103,104,105]. To address this point, Atianand et al. genetically deleted the entire 4-kb genomic locus harboring the lncRNA Ttc39a opposite strand RNA 1 (*Ttc39aos1*, also known as *lincRNA-EPS*) to generate *lincRNA-EPS*^−/−^ mice [106]. Using macrophages isolated from *lincRNA-EPS*-deficient mice, the authors uncovered that *lincRNA-EPS* interacts with heterogeneous nuclear ribonucleoprotein L (hnRNPL) to suppress the transcription of immune response genes (i.e., interferon-induced protein with tetratricopeptide repeats 2 (*Ifit2*) and radical S-adenosyl methionine domain containing 2 (*Rsad2*)).

Besides the transcriptional control above, the post-transcriptional regulation of TLR signaling via lncRNAs are reported. The most prominent ones are those investigating ceRNAs/miRNA sponges to sequester miRNAs to interfere with mRNA degradation. For example, the imprinting lncRNA, Maternally Expressed Gene 3 (*Meg3*; more specifically, *MEG3-4* transcript), binds to *miR-138*, which targets the proinflammatory cytokine *IL-1β*. Another example is *NEAT1*, which binds to *miR-15a* [107] and *miR-17-5p* [108] to control the translation of TLR4. Taken together, lncRNAs play active roles in regulating TLR signaling in macrophages.

NF-κB signaling is one of the most studied immune pathways since its discovery in 1986 [109]. It consists of five subunits (NF-κB1 p50, NF-κB2 p52, RelA/p65, RelB, and cREel) that can dimerize to form unique transcription factors that can either interact with other gene promoters via κB consensus motifs or with five inhibitory proteins that comprise the IκB family [110]. IκB binding to NF-κB dimers inhibits translocation to the nucleus by occluding DNA-binding Rel proteins and masking the nuclear localization signal region of the NF-κB subunit. The diversity of NF-κB activation is vast, and canonical and noncanonical signaling pathways have been reviewed extensively elsewhere [110,111,112]. NF-κB signaling in macrophages largely follows the same pattern of regulation, albeit mostly on the activation of pattern recognition receptors like TLRs and a heavier reliance on c-Rel as a vital component for the activity of NF-κB dimers [110,112]. NF-κB dimers are henceforth uninhibited by their release from the inhibitory IκB subunit and can be translocated into the nucleus, where they function as a transcription factor for a plethora of inflammatory genes, such as those that make up the well-known NOD-, LRR- and pyrin domain-containing protein 3 (NLRP3) inflammasome, a hallmark of the M1 phenotype [113].

LncRNAs have been found to both upregulate and suppress immune-responsive genes. Several of these lncRNAs have now substantiated roles in the regulation of NF-κB signaling in many human disease models in mice, including CVD [111,114,115,116,117,118,119,120,121,122,123,124]. The crosstalk between lncRNAs and NF-κB signaling is complex, involving both direct and indirect mechanisms and spanning from epigenetic control and even to post-translational modifications. The expression of long intergenic non-coding RNA (lincRNA) prostaglandin-endoperoxide synthase 2, opposite strand 2 (Ptgs2os2, also known as *lncRNA-Cox2*), is known to be highly induced in macrophages upon TLR activation. Hu et al. [125] found that *lncRNA-Cox2* is directly involved in modulating the assembly of NF-κB subunits to the SWI/SNF complex, thereby acting as an epigenetic coactivator of the NF-κB-mediated late-primary response gene transcription in murine macrophage (microglia and RAW264.7) cells [125]. Further evidence for chromatin remodeling via lncRNAs was shown in a study where *lincRNA-Tnfaip3* was found in the LPS-stimulated macrophages to regulate high-mobility group box 1 (Hmgb1)-associated histone modification and the subsequent NF-κB-mediated transactivation of pro-inflammatory genes [126]. Other studies in murine bone marrow-derived macrophages demonstrated that the expression of the lncRNA cardiac and apoptosis-related lncRNA (*Carlr*) is not only directly increased by NF-κB activation but, more importantly, its knockdown results in significantly reduced expression levels of several NF-κB-regulated genes, such as the nuclear factor of the kappa light polypeptide gene enhancer in B cells 2, p49/p100 (*Nfkb2*), prostaglandin-endoperoxide synthase 2 (*Ptgs2*), interleukin 1 alpha (*Il1a*), and interleukin 1 beta (*Il1b*) [127].

The lncRNA Firre intergenic repeating RNA element (*FIRRE*) is a species-conserved lncRNA (between humans and mice) that is upregulated by NF-κB signaling. Lu et al. [128] showed an interesting post-transcriptional mechanism whereby *FIRRE* directly interacts with heterogeneous nuclear ribonucleoproteins (hnRNPs) to stabilize the mRNAs of inflammatory genes (*TNF-a*, *IL1b*, and *IL6*) following LPS stimulation. Similarly, the post-translational modulation of inflammatory cytokines was shown in microglial cells whereby the lncRNA-predicted gene 4419 (*Gm4419*) was found to be upregulated and promoted the phosphorylation of the inhibitor of nuclear factor kappa B (IκBα) by direct physical association, ultimately leading to the increased transcriptional activation of *TNF-α*, *Il1b*, and *Il6* [129]. The aforementioned *FIRRE* has also been shown to form a positive feedback loop to promote the NLR family pyrin domain containing 3 (NLRP3) inflammasome formation in microglial cells [130]. Adding to the complexity of lncRNAs’ roles in inflammatory diseases, the lncRNA *HIX003209* was shown to promote macrophage activation in THP-1 cells by acting as a ceRNA for *miR-6089* through a TLR4/IκB/NF-κB signaling pathway [131]. The lncRNA *Lethe* also appears to be important in metabolic diseases, such as diabetes, by regulating the ROS production in RAW264.7 macrophages via NF-κB signaling [132].

The indirect regulation of macrophage functions by lncRNAs and NF-κB crosstalk has also been shown in mice peritoneal macrophages whereby the lncRNA myocardial infarction-associated transcript 2 (*Mirt2*) acts as a molecular checkpoint to prevent the aberrant activation of a proinflammatory program that potently regulates the macrophage phenotype. *Mirt2* is induced in the LPS-stimulated macrophages and inhibits the K-63 ubiquitination of TNF receptor-associated factor 6 (TRAF6) and, thus, attenuates NF-κB signaling via negative feedback. *Mirt2* knockdown suppressed the proinflammatory program in LPS-stimulated macrophages [133]. Together, lncRNAs are involved in the regulation of NF-kB functions in macrophages. Increasing our understanding of how the complex lncRNA network interplays with NF-kB signaling would provide new potential molecular targets to limit the proinflammatory activation of macrophages in the pathophysiology of CVD. 

## 7. Conclusions

Recent studies suggest that lncRNAs play a crucial role in the regulation of myeloid recruitment and lineage determination (e.g., the polarization of macrophages to proinflammatory M1 and reparative M2 macrophages), as well as innate and adaptive immune functions. Although significant progress has been made in macrophage biology, little is known regarding lncRNAs in the relationship between macrophages and failing hearts. Furthermore, due to the lack of species conservation of lncRNAs (e.g., between humans and mice), the studies of lncRNAs in model organisms are not well-received in human diseases, including CVD. Adding to this lack of species conservation, most of the recent lncRNA studies fail to provide the detailed characteristics of lncRNAs, including the presence of isoforms and other non-coding RNAs (ncRNAs). For example, one of the most well-studied lncRNA, *MEG3*, owns 50 human isoforms (Ensemble database accession ENSG00000214548). Within its locus, several other ncRNAs are encoded (i.e., *AL117190.1-201*, *AL117190.3-201*, *AL117190.7-201*, and *MIR770-201*), which raises a significant caution in interpreting the loss-of-function experimental results. In the case of *MEG3*, gain-of-function experiments will be difficult to conduct if all 50 *MEG3* isoforms are to be cloned and overexpressed in experimental settings. However, it is imperative to conduct more careful studies of the lncRNA locus itself, as well as the validation of isoforms via the rapid amplification of cDNA ends (RACE) and Northern blotting assays to move the lncRNA field forward to avoid the misinterpretation of experimental results due to ambiguity surrounding the functionalities of lncRNAs.

## Figures and Tables

**Figure 1 ncrna-06-00028-f001:**
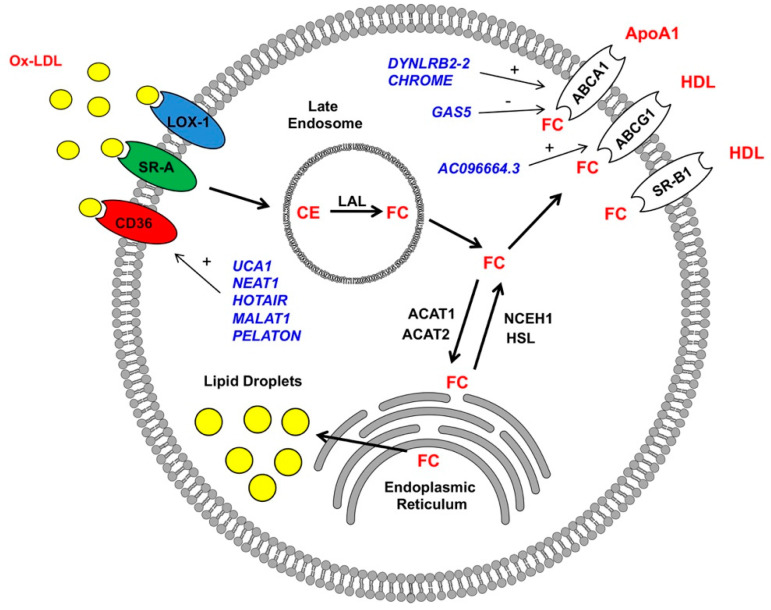
Long non-coding RNAs (LncRNAs) regulate macrophage-derived foam cell formations. LncRNAs (blue) have positive (+) and negative (-) impacts on the expressions of scavenger receptor CD36 and reverse cholesterol transporters ABCA1 and ABCG1 in macrophages (details in text). Ox-LDL: oxidized low-density lipoproteins, HDL: high-density lipoproteins, FC: free cholesterol, LAL: lysosomal acid lipase, and CE: cholesteryl esters.

**Figure 2 ncrna-06-00028-f002:**
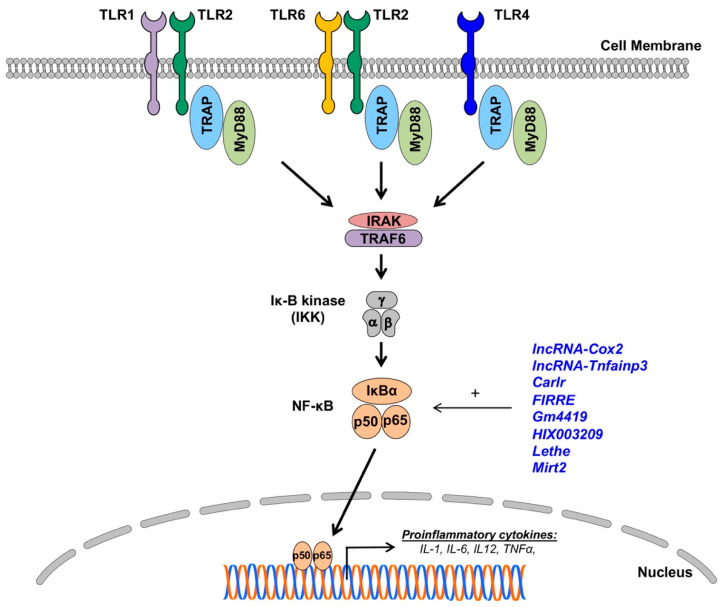
The impact of lncRNAs on Toll-like receptor (TLR) expressions and nuclear factor (NF)-κB signaling in macrophages. LncRNAs (blue) enhance (+) NF-κB signaling and proinflammatory gene expressions in macrophages (details in text).

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
