# Peer review of "Macrophage Long Non-Coding RNAs in Pathogenesis of Cardiovascular Disease"

_ncrna, 2020, doi:10.3390/ncrna6030028_

Round 1

Reviewer 1 Report

This is a very interesting review. A few comments arose.

1. In my opinion, the authors need to discuss in more detail the origin of macrophages of the heart and blood vessels. There are several basic studies and reviews on this subject.

Hoeffel, G. & Ginhoux, F. Fetal monocytes and the origins of tissue-resident macrophages. Cell. Immunol. 330, 5–15 (2018).

Perdiguero, E. G. & Geissmann, F. The development and maintenance of resident macrophages. Nat. Immunol. 17, 2–8 (2016).

2. It may be worthwhile to structure and summarize data on the role of lncRNA in a particular pathology and present in a table.
3. Is there evidence of the use of lncRNA as therapeutic targets?

Author Response

Reviewer #1

This is a very interesting review. A few comments arose.

Response: Thank you very much for your praise.

  1. In my opinion, the authors need to discuss in more detail the origin of macrophages of the heart and blood vessels. There are several basic studies and reviews on this subject.

Hoeffel, G. & Ginhoux, F. Fetal monocytes and the origins of tissue-resident macrophages. Cell. Immunol. 330, 5–15 (2018).

Perdiguero, E. G. & Geissmann, F. The development and maintenance of resident macrophages. Nat. Immunol. 17, 2–8 (2016).

Response: We have modified the Introduction section to describe the origin of tissue-resident macrophages by citing the articles that were suggested.

  1. It may be worthwhile to structure and summarize data on the role of lncRNA in a particular pathology and present in a table.

Response: Although it is a great suggestion to categorize lncRNAs according to different pathologies of cardiovascular disease, most of the studies published so far are based on experimental results in the cell culture systems. The main objective of this review manuscript is to introduce the readers to the current status of lncRNAs in macrophages to further stimulate the research into cardiovascular disease and its different etiologies, especially using model organisms (e.g., mice, rats) and using cardiovascular disease patients' samples. Because of these reasons, we are unable to generate such table due to lack of sufficient number of publications providing mechanistic insights of lncRNA regulation in macrophages related to cardiovascular disease.

  1. Is there evidence of the use of lncRNA as therapeutic targets?

Response: Despite the high hope for lncRNAs as therapeutic targets, only very few lncRNAs have been approved by the Food and Drug Administration (FDA) at the moment due to lack of extensive validation studies. For example, the lncRNA prostate cancer antigen 3 (PCA3, also known as DD3) is being used as a biomarker in urine for prostate cancer diagnosis.

Reviewer 2 Report

In this manuscript, the authors have systematically summarized the current findings on the relationships between lncRNA and CVD. Since there are a few similar review articles that were published in less than a year ago, it would be helpful to both incomers and experienced researchers in different areas if there is a brief description on the differences between the contents of this manuscript and the other recent review articles, such as the following two examples.

Vascul Pharmacol. 2019 Mar;114:23-30.

Curr Opin Cardiol. 2019 May;34(3):241-245.

Author Response

Reviewer #2

In this manuscript, the authors have systematically summarized the current findings on the relationships between lncRNA and CVD. Since there are a few similar review articles that were published in less than a year ago, it would be helpful to both incomers and experienced researchers in different areas if there is a brief description on the differences between the contents of this manuscript and the other recent review articles, such as the following two examples.

Vascul Pharmacol. 2019 Mar;114:23-30.

Curr Opin Cardiol. 2019 May;34(3):241-245.

Response: Thank you very much for valuable comments. We added the suggested recent review articles and the following new sentence at the end of Introduction section:

“Compared to other similar review articles in recent years [12,13], we summarize current knowledge regarding lncRNAs in directing various aspects of macrophage function in CVD, including foam cell formation, toll like receptor (TLR) and NF-kβ signaling, and macrophage phenotype switching.”

Round 2

Reviewer 1 Report

Thanks for the replies and corrections made to the manuscript.